# An automated high-content screening and assay platform for the analysis of spheroids at subcellular resolution

**Margaritha M. Mysior** [ID][◔], **Jeremy C. Simpson** [ID] *[◔]

Cell Screening Laboratory, UCD School of Biology & Environmental Science, University College Dublin, Dublin, Ireland

◔ These authors contributed equally to this work.
* jeremy.simpson@ucd.ie

## Abstract

The endomembrane system is essential for healthy cell function, with the various compartments carrying out a large number of specific biochemical reactions. To date, almost all of our understanding of the endomembrane system has come from the study of cultured cells growing as monolayers. However, monolayer-grown cells only poorly represent the environment encountered by cells in the human body. As a first step to address this disparity, we have developed a platform that allows us to investigate and quantify changes to the endomembrane system in three-dimensional (3D) cell models, in an automated and highly systematic manner. HeLa Kyoto cells were grown on custom-designed micropatterned 96-well plates to facilitate spheroid assembly in the form of highly uniform arrays. Fully automated high-content confocal imaging and analysis were then carried out, allowing us to measure various spheroid-, cellular- and subcellular-level parameters relating to size and morphology. Using two drugs known to perturb endomembrane function, we demonstrate that cell-based assays can be carried out in these spheroids, and that changes to the Golgi apparatus and endosomes can be quantified from individual cells within the spheroids. We also show that image texture measurements are useful tools to discriminate cellular phenotypes. The automated platform that we show here has the potential to be scaled up, thereby allowing large-scale robust screening to be carried out in 3D cell models.

## Introduction

The human body consists of millions of cells that are organised into tissues and organs allowing them to carry out specialised functions. Central to this functionality is that cells have the ability to communicate with one another via signalling networks. At one level, this is achieved by cells modulating the contents of their plasma membrane. This occurs via an extensive set of subcellular trafficking pathways, which serve to both internalise cell surface molecules as well as deliver newly synthesised molecules to the plasma membrane in a timely manner. Membrane trafficking pathways between the various subcellular organelles have been extensively

BioImage Archive (https://www.ebi.ac.uk/biostudies/bioimages/studies) at DOI:10.6019/S-BIAD1259.

**Funding:** MMM is supported by the Government of Ireland Postdoctoral Fellowship from the Irish Research Council (GOIPD/2023/1135). The funders had no role in study design, data collection and analysis, decision to publish, or preparation of the manuscript."

**Competing interests:** The authors have declared that no competing interests exist.

studied, and their regulation occurs via a wide variety of molecular machinery [1–3]. Tight control of compartment identity and membrane flux throughout the cell allows it to precisely orchestrate various functions, including protein, lipid and carbohydrate synthesis as required, the sorting and recycling of plasma membrane proteins, and extracellular interactions.

Much of our knowledge of the endomembrane system has come from cultured mammalian cells grown as monolayers. Although these two-dimensional (2D) grown cells are easy to grow and manipulate, they do not resemble the physiological organisation of cells as found in tissues. Furthermore, it remains unclear as to whether their intracellular organisation, when grown as monolayers, reflects the situation found in tissues. In the last twenty years, three-dimensional (3D) models have emerged as an additional tool with which to study cell function. Cancer research, drug delivery, and toxicology studies are increasingly utilising 3D cell models to better recapitulate cell physiology [4–6]. Laboratory-generated 3D cell models can broadly be categorised into either spheroids or organoids. Spheroids are cell assemblies, and many commercially available cell lines are used to grow spheroids. In the case of cancer cell lines, spheroids are called multicellular cancer tumour spheroids (MCTS) [5,7]. When stem cells are used to grow 3D cell models, they are called organoids, and if grown under certain conditions they can differentiate to generate 3D assemblies containing multiple cell types [8–10]. Both spheroids and organoids offer huge potential to the research community, depending on the biological question under study. While organoids potentially provide greater similarity to tissues, spheroids are easier and faster to culture and are more amenable to perturbation using established molecular genetic tools.

There are typically two ways to grow 3D cell models, either using or not using scaffolds to facilitate formation of the 3D assembly. These methods are reviewed in detail elsewhere [11–14]. The scaffolds that are used can be made from various materials, including polymers that are from synthetic or natural sources such as hydrogels or extracellular matrix extract [15,16]. By contrast, scaffold-free methods use a non-adhesive surface coating to prevent cells from adhering to the surface, forcing them to form aggregates [11–14]. Sometimes, a combination of both methods is used, such as imprinting microwells into a hydrogel that has non-adhesive properties stopping the cells from adhering to the microwells [17,18].

Despite the promise that 3D cell models offer, and indeed the growing number of calls for the study of fundamental cellular processes in 3D [19–21], to date there are few studies that have utilised them in this context. This is surprising given that it has been shown that 3D cell assemblies can have altered gene expression and drug response patterns compared to monolayer-grown cells [4,22–26]. This raises questions about whether all our current knowledge of cells obtained from monolayer cultures can be extrapolated to explain how cells work in tissues and organs. The first reported use of a simple 3D cell model to assess intracellular trafficking pathways was reported by Mrozowska and Fukuda in 2016 [27]. They grew Madin-Darby Canine Kidney (MDCK) II epithelial cells as 3D 'cysts' in Matrigel, an extracellular matrix extract, and visualised the single-layered hollow cysts by fluorescence microscopy. They investigated the distribution of podocalyxin, an apical membrane marker, in the context of the Rab family of small GTPases, and compared their findings to that seen in traditional 2D cell monolayer cells. The study highlighted that the depletion of a number of Rab proteins resulted in different phenotypes in monolayer versus 3D cysts. Furthermore, they also highlighted that Rab35 interacts with different effectors in 2D versus 3D grown cells to regulate the transport of podocalyxin. More recent work from the same laboratory has shown that Rab35 has two distinct guanine nucleotide exchange factors in 2D and 3D, namely DENN (differentially expressed in normal cells and neoplasia) 1A (DENN) and folliculin, respectively [28]. Other work from the Fukuda lab showed differences in cyst formation of Rab knock-out cell lines. Of interest was the Rab6 knock-out cyst that did not produce a basement membrane similar to

that seen with other Rab knock-out cysts. These changes in the basement membrane were not evident in monolayer cells [29]. Together, these studies emphasise that potentially not only are there gene expression differences between 2D and 3D cell models, but also changes to the sub-cellular localisation of proteins and patterns of membrane organisation. Undoubtedly there are significant challenges to working with 3D cell assemblies, not least of which are uniformity of the assembly and the ability to be able to resolve and quantify morphological changes at the individual cell and even subcellular level. Nevertheless, given the important observations described above, and the increasing interest in using spheroids and organoids, it seems timely to explore whether the function of the endomembrane system can be explored in multicellular models. To address this, here we describe a platform that allows the routine production of large numbers of highly uniform solid spheroids, which can be imaged using automated high-content screening microscopy. Our approach uses micropatterning within 96-well plates [30], allowing the generation of large numbers of arrayed spheroids. We utilise this platform to make single-cell and subcellular measurements on various membrane organelles in the context of drug perturbations, and show that a classical membrane traffic assay can also be carried out and quantified from individual cells within a 3D cell model.

## Materials and methods

### Cell culture

HeLa Kyoto cells (human cervical cancer cell line, RRID;CVCL-1922) were routinely cultured in Dulbecco's modified Eagle medium (DMEM) (Life Technologies) with 1 g/l glucose supple-mented with 10% heat-inactivated foetal bovine serum (FBS) (PAA Laboratories) and 1% glu-tamine (Life Technologies) in 10 cm Nunclon culture dishes (Thermo Scientific) at 37˚C in a humidified atmosphere of 5% CO/ 95% air. Cells were subcultured (1:10 dilution) upon reach-ing confluency by first rinsing the dish with a 0.05% trypsin-EDTA solution (Life Technolo-gies) to remove dead cell debris and remaining growth medium, followed by incubation at 37˚C until cells detached. Cells were used up to passage 15, after which they were discarded. HeLa Kyoto cells stably expressing EGFP, EGFP-Rab5A, or EGFP-Rab6A were cultured in the parental cell complete medium supplemented with 700 μg/ml G418 sulphate (Life Technolo-gies). Stable cell lines were produced by transfection of HeLa Kyoto cells with relevant DNA plasmids encoding the GFP-fusions of interest. After 24 h, the medium was exchanged with medium containing 700 μg/ml G418 sulphate. Cells were grown for a further two weeks in this medium. Distinct cell colonies were isolated following trypsin treatment, and passed through 50 μm CellTrics filters (Sysmex) and sorted by a BD FACSAria III Cell Sorter (BD Biosci-ences). Single EGFP-positive cells displaying a low-medium fluorescence signal were sorted into individual wells of a CellCarrier-96 Ultra microplate (Revvity) and then propagated.

### Cell culture in micropatterned plates

Cells were plated into custom designed micropatterned plates (CYTOO SA) with a disc size of 45 μm and a pitch between the micropatterns of 300 μm (DC45-P300-FN) to enable spheroid growth. The disc micropattern was coated with fibronectin by the supplier. Prior to use, the plate was warmed to room temperature for 1 h, followed by addition of 100 μl complete medium into the wells, and further incubation at 37˚C for 1h. After this, 5 000 cells per well in 100 μl complete medium were plated into the wells and grown for 3 days to allow the forma-tion of spheroids. Cells were plated using either a single channel or 8-channel pipette, depend-ing on the number of wells needed per plate.

## Immunofluorescence of spheroids

Spheroids grown on CYTOO micropatterned plates were fixed with 100 μl of 3% PFA (Sigma) for 1 h at room temperature, followed by quenching with 100 μl of 30 mM glycine (Fisher Scientific) in PBS for 30 min and two PBS (Sigma) washes. Spheroids were permeabilised with 100 μl of 0.5% Triton X-100 (Sigma) in PBS for 1 h at room temperature. Primary antibodies (see Table 1) and secondary antibodies (see Table 1) containing 0.2 μg/ml Hoechst 33342 (Sigma) were diluted in PBS containing 0.02% sodium azide (Sigma) and incubated with the spheroids for 2 h at RT. During the optimisation phase, several other antibodies were tested at a range of dilutions. Typically 100 μl solution per well was used. The addition and removal of liquid (antibodies, PBS, other solutions used for processing the spheroids) was carried out using either a multichannel pipette or a semi-automated pipetting robot (Integra ViaFlo-96) set on the slowest aspiration and dispensing speeds.

## Drug assays

Inhibitors of membrane trafficking events were used to invoke changes to the Golgi apparatus. For this, HeLa Kyoto, EGFP-HeLa Kyoto, EGFP-Rab5A HeLa Kyoto and EGFP-Rab6A HeLa Kyoto cells were seeded into the customised CYTOO DC45-P300-FN plates. On the final day of growth, cells were treated with brefeldin A (BFA, 10 μg/ml) (Sigma) or nocodazole (7.5 μM) (Sigma) for 30 min at 37°C. Subsequently, spheroids were fixed and stained with markers of the Golgi (GM130), as described above.

## Image acquisition and analysis

All images were acquired with an Opera Phenix High-Content Screening System (Revvity) using various water immersion objectives (20x/1.0 NA and 63x/1.15 NA). The 'PreciScan' feature of the Opera Phenix High Content Screening System was used to pre-scan each well with a low magnification objective for objects of interest (spheroids) and then a selected number of these objects were reimaged with a high magnification objective. In general, 49 fields of view (a tile of 7x7 images) were imaged for the pre-scan, which is equivalent to ca. 68% of the entire well using the 20x objective lens. Typically, 65 planes, each 1 μm apart, were acquired. This step size was selected as it facilitated a reasonable total imaging time across the plate (see below) and manageable data size for analysis and storage, both of which are considerations for screening approaches. 100% laser power and exposure times between 120 ms and 500 ms were used. Imaging times for the pre-scan were 45–60 min for 60 wells, and for the high magnification rescan, they were 6–9 h, depending on the number of spheroids identified per well and the number of wells imaged. We did not experience any issues with photobleaching during the imaging of the spheroids with these exposure times used.

**Table 1. Antibodies used.**

| Antibody | Dilution | Company |
|---|---|---|
| Rb-α-GM130 (D6B1) | 1:500 | Cell Signalling Technologies |
| Mo-α-EEA1 (clone 14/EEA1) | 1:200 | BD Transduction Laboratories |
| Mo-α-LAMP1 (H4A3) | 1:200 | Developmental Studies Hybridoma Bank |
| Alexa Fluor® 568 go-α-Rb IgG (H+L)—highly cross-adsorbed | 1:400 | Thermo Fisher Scientific |
| Alexa Fluor® 647 go-α-Mo IgG (H+L)—highly cross-adsorbed | 1:400 | Thermo Fisher Scientific |

Images were analysed using Harmony v4.8 software (Revvity). The 3D analysis routines within Harmony calculate a 3D image based on the selected z-planes and the entire 3D image is analysed. The spheroid region was first identified and segmented based on the Alexa Fluor 647 signal before the other steps were carried out. This segmentation can be also carried out using plasma membrane staining (if used) or the Hoechst 33342 staining. The nuclei were segmented based on the Hoechst 33342 signal, followed by segmentation of the cytoplasm based on the residual Hoechst 33342 signal detected in the cytoplasm. For cells stably expressing the various EGFP-tagged proteins, the intensity of the EGFP signal of the cells was measured, and cells deemed not expressing EGFP were discarded from the analysis. Next, the organelles were identified and segmented. For large distinct structures, such as the Golgi apparatus, the most effective segmentation was achieved by applying a Gaussian filter. For smaller organelles, such as endosomes and lysosomes, a texture PLS (Plane, Line, Saddle) Spot Bright filter was applied. A volumetric analysis was then performed, and features such as volume, sphericity, number of fragments were calculated for the Golgi, early endosomes, and lysosomes. For drug-treated spheroids, texture features were used to describe the redistribution of the EGFP-Rab6A signal in cells. The PLS texture features 'plane bright' and 'saddle' are the volumetric equivalent to SER (Saddle, Edge, Ridge) texture features, commonly used for single image plane analysis of monolayer cells. Detailed image analysis methods can be found in S1–S7 Tables in the Appendix. Graphs were assembled in R studio v4.3.2. For creating axis breaks for some graphs, the package ggbreak [31] was used. Figures were assembled in Adobe Photoshop.

## Statistical analysis

Statistical analysis was performed using R Studio v4.3.2. An Anova test with post-hoc Tukey HSD was used to detect differences of spheroid measurements. A one-sample Kolmogorov-Smirnov test was performed to check the normal distribution of the single cell data. As the data were not normally distributed an unpaired Wilcoxon test was performed.

## Results

Despite the increasing realisation that 3D cell models need to be utilised more frequently in fundamental life science research, their deployment is still very limited. Similarly, 3D cell models offer exciting possibilities for a deeper understanding of the effects of therapeutics on cells. The majority of researchers using 3D cell models grow them in hydrogel/matrix-based systems [27,28] or use ULA methods [32,33]. These methods either produce several spheroids per well, but the spheroids are not uniform in size and shape; or in the case of ULA approaches, uniform spheroids are produced, but typically only one spheroid per well. To overcome both of these critical limitations, we designed an approach that would facilitate the growth of several hundred highly uniform spheroids per well, and which would be compatible with fully-automated high-resolution fluorescence imaging such that we could study the endomembrane system within individual cells within the 3D assembly. To do this, we used commercially manufactured micropatterned plates in which we custom-designed the pattern diameter and pitch to best allow adherence of sufficient cells to act as a seed for spheroid formation. Following optimisation (not shown), we determined that a disc pattern diameter of 45 μm and a pitch of 300 μm were ideal parameters for our purposes. We tested plating various numbers of cells (3 000–20 000 cells/well), increasing times of cell incubation (3–7 days) and different cell types (HeLa Kyoto and HEK293 cells). Additional cell lines capable of growing on micropatterned plates have been previously tested by Monjaret and colleagues [30]. Of the two cell lines tested in our experiments, HeLa Kyoto cells formed spheroids much more consistently than HEK293 cells, hence these were used for the remainder of the study. These plates allowed us to

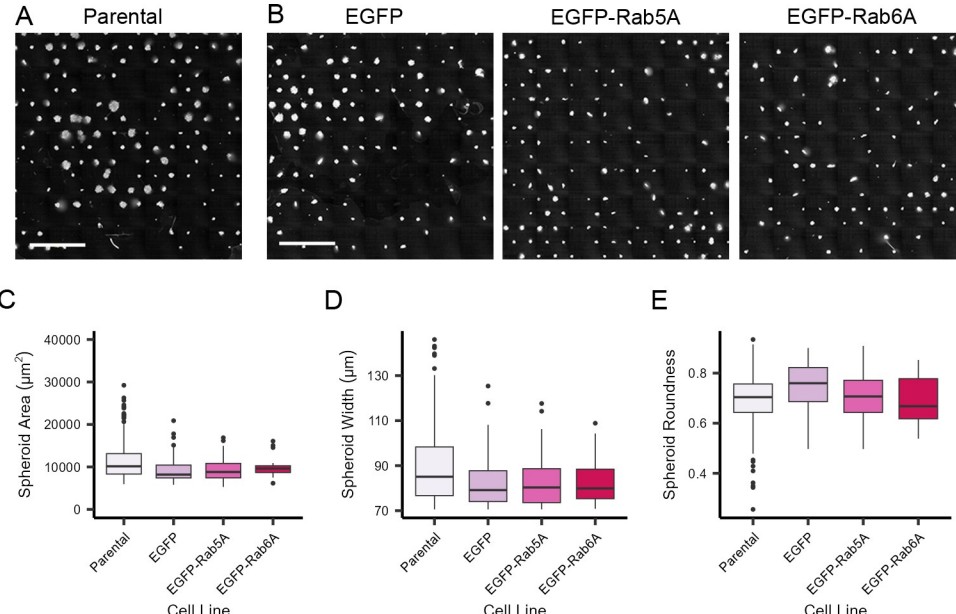

**Fig 1. Primary characterisation of spheroids grown on micropatterned plates.** Cells were grown on a CYTOO DC45-P300-FN plate for three days. Spheroids were fixed and stained with Hoechst 33342 (nuclei). Images were acquired with an Opera Phenix High-Content Screening microscope with a 20x/1.0 NA water immersion objective. A) Paternal (non-transfected) HeLa Kyoto spheroids. B) HeLa Kyoto spheroids stably expressing various constructs as indicated. Scale bars: 1 mm. Images are stitched from a 7x7 grid of single fields of view. C) Quantitative measurement of spheroid cross-sectional area for all four cell lines. D) Quantitative measurement of spheroid width. E) Quantitative measurement of spheroid roundness. The box plots show per spheroid measurements indicating the median values and 50 percentiles of the data. Number of spheroids analysed were 344 (parental), 89 (EGFP), 55 (EGFP-Rab5A), and 17 (EGFP-Rab6A).

produce several hundred HeLa Kyoto cell spheroids per well that were uniform in size and shape just 3 days after seeding, visualised by staining of the cell nuclei (Fig 1A). The fraction of successful cell occupancy on the micropatterns was ca. 90% shortly after seeding, and after 3 days of growth ca. 60–80% of the micropatterns supported spheroid formation. In order to test the wider applicability of this system for the study of the endomembrane system in cells, and to overcome the potential problem of uneven transient transfection in the 3D model, we generated a series of cell lines stably expressing relevant markers of the endomembrane system tagged with EGFP, including the small GTPases Rab5A (associated with early endosomes) and Rab6A (associated with the Golgi apparatus). A stable cell line expressing EGFP only was used as a control. These stable cell lines were seeded on the micropatterned plates, and although they did not form spheroids on every micropatterned position within each well, the efficiency of spheroid production was visually similar to that seen with the parental cells (Fig 1B). Spheroids from parental HeLa Kyoto cells and EGFP, EGFP-Rab5A and EGFP-Rab6A stably expressing HeLa Kyoto cells were imaged and then analysed using automated high content imaging and analysis, allowing single plane measurements of area, width and roundness to be measured from several spheroids in each well (Fig 1C–1E). This analysis revealed both a high degree of homogeneity across each population of spheroids per well, and also between the different cell lines. Typically the coefficient of variance (CV) between the number of spheroids identified between replicate wells was in the order of 5% for the same spheroid type.

One purpose of developing a spheroid model was to facilitate the study of individual cells within the context of their 3D environment. Therefore, using automated imaging with a high numerical aperture objective lens, we acquired image stacks through each spheroid, typically

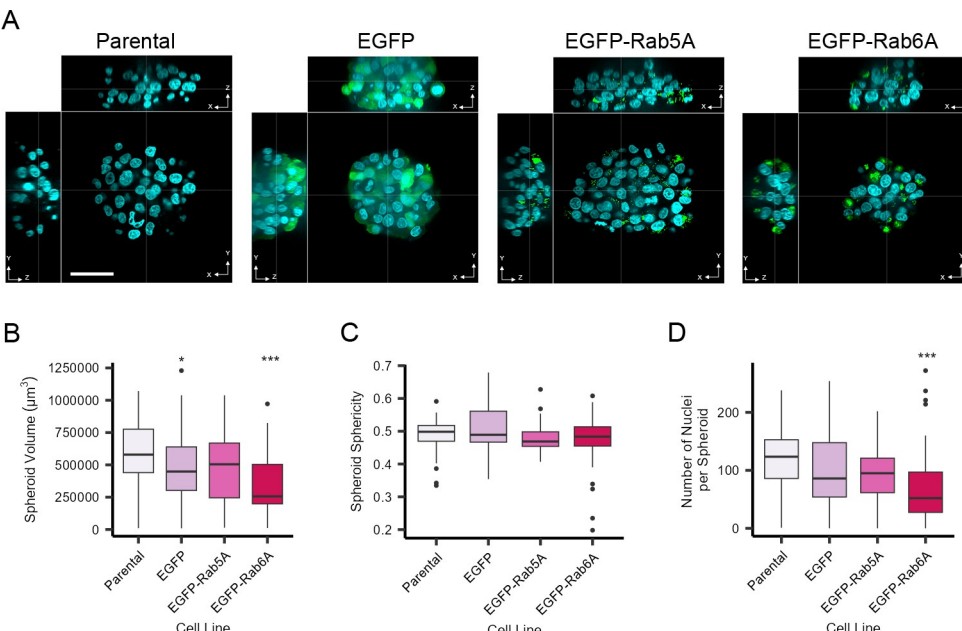

**Fig 2. Volumetric characterisation of spheroids grown on micropatterned plates.** Cells were grown on a CYTOO DC45-P300-FN plate for three days. Spheroids were fixed and stained with Hoechst 33342 (nuclei). Images were acquired with an Opera Phenix High-Content Screening microscope with a 63x/1.15 NA water immersion objective. A) XYZ views of spheroids from various cell lines as indicated. Aqua represents the nuclei and green the EGFP signal. Scale bar: 50 μm. B) Quantitative measurement of spheroid volume for all four cell lines. C) Quantitative measurement of spheroid sphericity for all four cell lines. D) Quantitative measurement of the number of nuclei per spheroid. The box plots show per spheroid measurements indicating the median values and 50 percentiles of the data. Number of spheroids analysed were 62 (parental), 56 (EGFP), 39 (EGFP-Rab5A), and 39 (EGFP-Rab6A). Data are from six technical replicate wells for each cell line. An Anova test with post-hoc Tukey HSD was carried out. * indicates p<0.05 and *** indicates p<0.001.

covering a range of 60–80 μm. This allowed us to make calculations with respect to spheroid volume (Fig 2B), shape (sphericity) (Fig 2C) and numbers of cells per assembly (Fig 2D). Although there was a relatively high degree of consistency between the different spheroids generated from the various cell lines, spheroids from EGFP-Rab6A cells were on average smaller than spheroids from parental cells (Fig 2B and 2D). Interestingly, the XYZ views of the spheroids (Fig 2A) and the spheroid sphericity values (Fig 2C) indicated that the spheroids adopted a more ellipsoid shape, with their height typically being approximately 50% of their width. The spheroids typically contained between 80 and 120 cells (Fig 2D), and at this size it was possible to segment the spheroids, nuclei, cytoplasm and organelles using automated image analysis software (S1 Fig).

We next assessed whether it was possible to also extract quantitative information at the subcellular level from the spheroids. We specifically decided to examine the Golgi apparatus, early endosomes, and lysosomes because these organelles play essential roles in various membrane trafficking pathways and changes to these organelles are linked to several human diseases [34–37]. Following growth of the spheroids for three days, as described above, we first immunostained them with antibodies against the Golgi matrix protein GM130. The Golgi apparatus was clearly visible in the individual cells in the spheroids generated from the various cell lines (Figs 3A–3D and S1). To gain deeper insight into the subcellular organisation of the cells growing in spheroid format, we again used automated confocal high-content imaging to acquire stacked images of the entire spheroid and apply volumetric analysis to each cell within the spheroid. Based on the GM130 immunostaining, this approach revealed that the nominal mean total

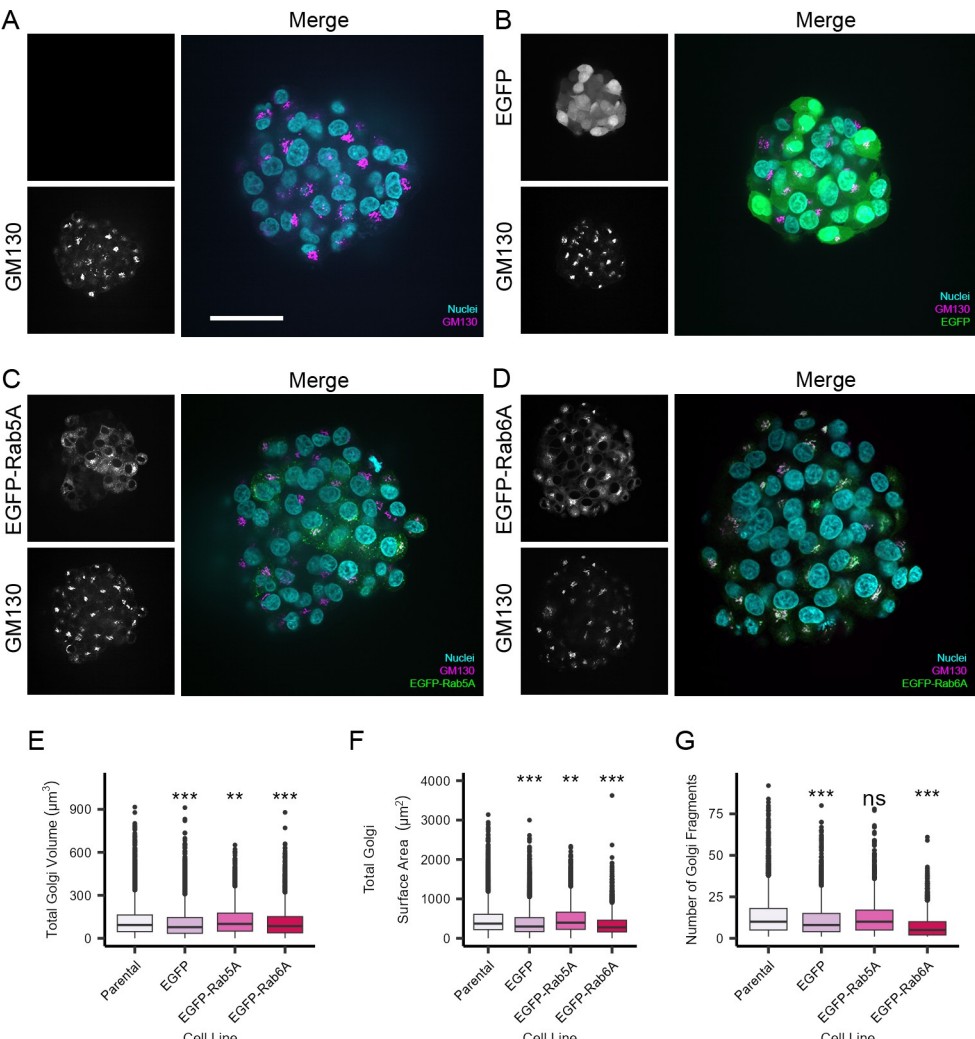

**Fig 3. Quantitative analysis of the Golgi apparatus in HeLa Kyoto cells growing as spheroids.** Cells were grown on a CYTOO DC45-P300-FN plate for three days to generate spheroids. The spheroids were fixed and stained with Hoechst 33342 (nuclei), and anti-GM130 antibodies were used to immunostain the Golgi apparatus. Images were acquired with an Opera Phenix High-Content Screening microscope using a 63x/1.15 NA water immersion objective. A-D) Representative images of the spheroids generated from the various cell lines. Scale bar: 50 μm. E) Mean total Golgi volume per cell. F) Mean total Golgi surface area per cell. G) Mean number of Golgi fragments per cell. The box plots show per cellmeasurements indicating the median values and 50 percentiles of the data. Number of cells analysed were 7053 (parental), 4939 (EGFP), 2068 (EGFP-Rab5A), and 2447 (EGFP-Rab6A) from a minimum of 39 spheroids per cell line. Data are from six technical replicate wells. An unpaired Wilcoxon test was carried out. ns indicates not significant, ** indicates p<0.01 and *** indicates p<0.001.

Golgi volume per cell, in cells grown as spheroids was ca. 125 μm$^3$ and that this showed good consistency across all four cell lines, irrespective of the GFP-tagged proteins that they were each expressing (Fig 3E). A similar trend was also seen in the total Golgi surface area per cell (Fig 3F). We next calculated the number of distinct Golgi objects in each cell. Although this organelle exists as a ribbon-like structure in cells, its state of fragmentation is not only intrinsically linked to a variety of functions but also various pathological situations [38]. Single-cell analysis revealed that in cells growing as spheroids, typically between 5 and 10 distinct fragments could be detected at the level of the fluorescence light microscope (Fig 3G).

We next repeated these experiments, but now immunostaining for early endosomes, represented by the early endosomal marker EEA1. Visual analysis of the images of the spheroids revealed that there was some variability of early endosomal distribution in cells within each spheroid, showing both dispersed and clustered patterns. However, there were no clear visual differences between the four cell lines used (S2A–S2D Fig). Volumetric analysis of the early endosomes determined that this organelle occupied a total volume of ca. 120 $\mu m^3$ per cell (S2E Fig), with a total surface area of 800 $\mu m^2$ (S2F Fig). In general, the different cell lines showed little difference in the number of endosomes, in absolute terms, that could be identified as distinct objects (S2G Fig). Finally, we immunostained the spheroids for acidic organelles, using antibodies against the lysosomal membrane protein LAMP1 (S3A–S3D Fig). Similar to the endosomes, we were able to measure the total cellular volume occupied by these organelles (S3E Fig). Interestingly, their calculated total volume was somewhat lower than that of the early endosomes, with a mean value of ca. 30 $\mu m^3$, likely because they displayed a highly clustered distribution, close to the nucleus. This was also reflected in the low values we obtained for numbers of lysosomes per cell (S3G Fig).

Having established that it was possible to automatically quantify a number of subcellular features within individual cells growing as spheroids, we next investigated whether changes to organelle morphology introduced by drug perturbation could also be quantified. We concentrated on the Golgi, an organelle where morphological changes can be more easily visualised, and where drugs are commonly available to induce morphological changes. We first selected brefeldin A (BFA), a metabolite that causes rapid redistribution of Golgi membranes into the endoplasmic reticulum (ER) within a few minutes of treatment. As shown in Fig 4, following treatment of the spheroids with 10 $\mu g$/ml BFA for 30 minutes, the Golgi apparatus (as defined by GM130 immunostaining), was seen to lose its typical juxtanuclear pattern. Instead, the GM130 was found in distinct punctate structures scattered throughout the cytoplasm of the cells within the spheroids (Fig 4B). This redistribution of GM130 appeared similar across all three cell lines examined. We applied some of the previously described quantification tools to the GM130 signal in the spheroids. Measurement of the total Golgi volume per cell revealed that after treatment with BFA there was a small and statistically significant reduction in mean volume of this organelle in each cell (Fig 4C). However, far more striking, was quantification of the number of detectable GM130 fragments, revealing a more than 20-fold increase in their number following BFA treatment (Fig 4D). The inclusion of the stably-expressing cell lines allowed us to examine other effects on the cells following BFA treatment. As expected, the early endosome population, judged from the EGFP-Rab5A signal, largely remained in a punctate distribution throughout the cell (Fig 4B), with only a small change in the mean number of structures (Fig 5A). By contrast, the EGFP-Rab6A pattern changed from being highly compact and co-localising with the GM130, to becoming web-like in appearance, reminiscent of the ER (Fig 4B). Quantification of the EGFP-Rab6A signal revealed an increase in the number of EGFP-Rab6A structures (Fig 5A). Given that the EGFP-Rab6A distribution changed following BFA treatment, we also explored other image analysis tools that might be able to quantify this effect. Measurement of EGFP-Rab6A footprint area within each cell was found to be a good discriminator, revealing that the EGFP-Rab6A fluorescence pattern showed on average a 20-fold increase in the BFA-treated cells compared to control cells (Fig 5B). Another useful set of tools in image analysis are texture features [39], which provide the opportunity to quantify complex patterns of fluorescence where simple counting or distance values cannot describe phenotypes. We found that the values that we obtained for two texture features in particular, namely 'plane bright' and 'saddle', also changed between the non-treated and treated cells (Fig 5B).

Finally, we wanted to test whether our approach had the potential to be applied to quantify other drug perturbations. Another compound known to cause morphological effects on the

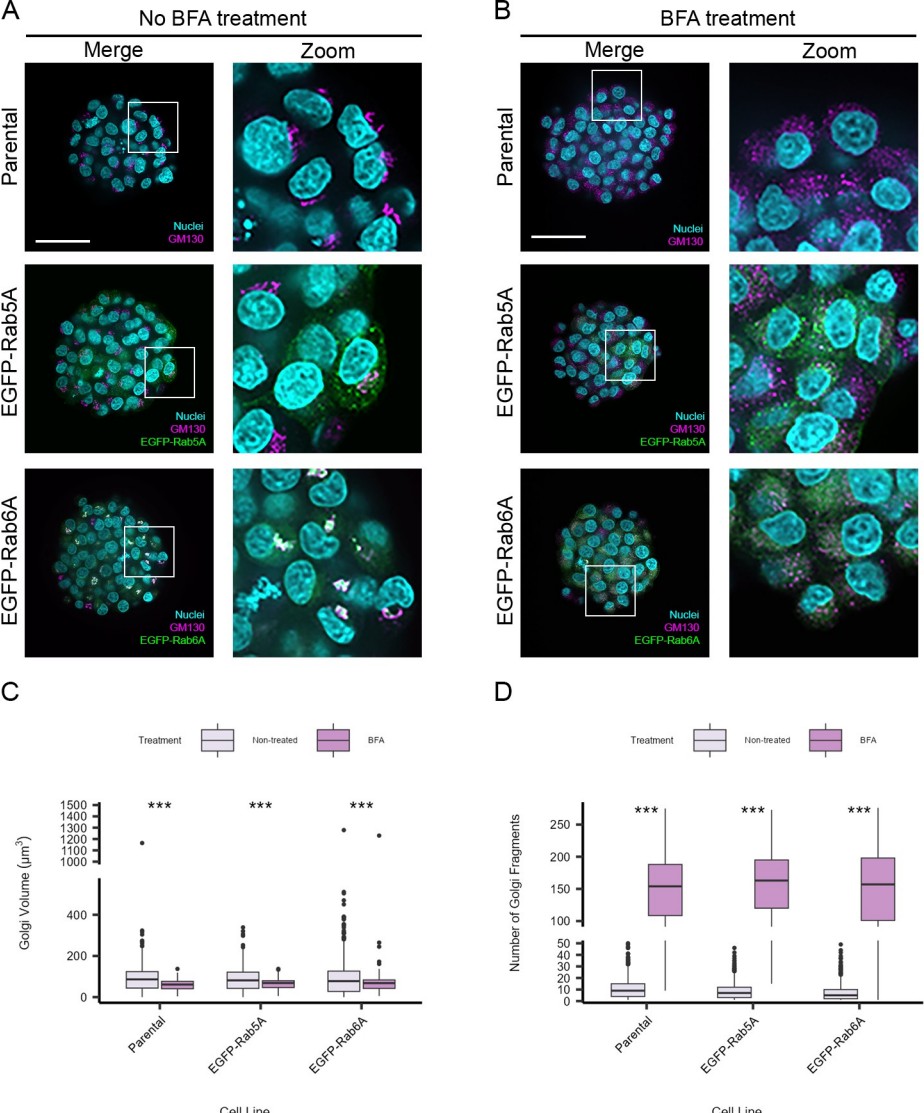

**Fig 4. BFA-induced Golgi perturbation in spheroids.** A) Representative images of untreated spheroids and B) BFA-treated spheroids, as indicated. After three days of growth in the CYTOO DC45-P300-FN plate, spheroids were treated with 10 μg/ml BFA for 30 min at 37˚C. Spheroids were fixed and stained with Hoechst 33342 (nuclei), and antibodies were used to immunostain GM130 (Golgi). Images were acquired with an Opera Phenix High-Content Screening microscope with the 63x/1.15 NA water immersion objective. Scale bars: 50 μm. After image acquisition, a volumetric analysis was performed. C) Quantification of Golgi (GM130) volume. D) Quantification of number of Golgi (GM130) fragments. The box plots represent per-cell measurements. Number of cells analysed were 953 (parental, non-treated), 935 (parental, BFA-treated), 758 (EGFP-Rab5A, non-treated), 831 (EGFP-Rab5A, BFA-treated), 845 (EGFP-Rab6A, non-treated) and 666 (EGFP-Rab6A, BFA-treated) from a minimum of 33 spheroids per cell line. Data are from two biological replicates. An unpaired Wilcoxon test was performed. *** indicates p<0.001.

Golgi apparatus is the microtubule depolymerising agent nocodazole. Spheroids were treated with 7.5 μM nocodazole for 30 minutes and their entire volumes were imaged using fully automated confocal microscopy. Nocodazole also caused fragmentation of the Golgi, albeit less visually pronounced than with BFA treatment (S4 Fig). Quantification revealed a small and statistically significant reduction in overall Golgi volume (S4C Fig), but as expected a significant increase in the number of detectable Golgi fragments, as judged from the GM130 immunostaining (S4D Fig). Similar to BFA treatment we could not detect changes in the number of

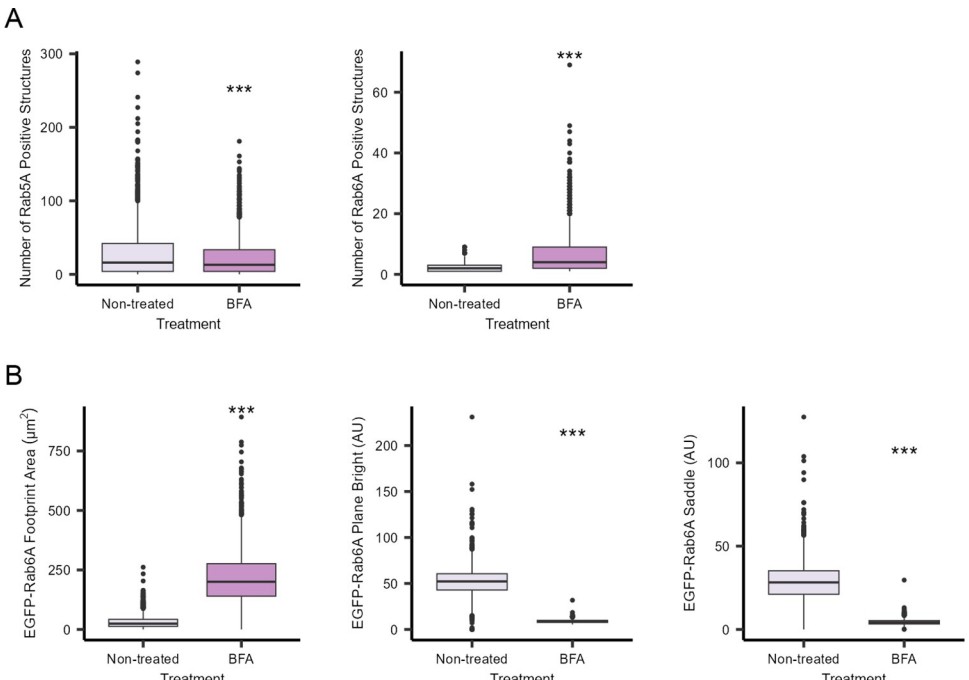

**Fig 5. BFA-induced EGFP-Rab5A and EGFP-Rab6A perturbation in spheroids.** After three days of growth in the CYTOO DC45-P300-FN plate, spheroids were treated with 10 μg/ml BFA for 30 min at 37°C. Spheroids were fixed and stained with Hoechst 33342 (nuclei), and antibodies were used to immunostain GM130 (Golgi). A) Number of EGFP-Rab5A and EGFP-Rab6A positive structures in untreated and BFA-treated cells. B) Quantification of EGFP-Rab6A morphometric (footprint area) and texture feature (plane bright and saddle) changes after BFA treatment. Number of cells analysed were 2016 (EGFP-Rab5A, non-treated), 2071 (EGFP-Rab5A, BFA-treated), 2066 (EGFP-Rab6A, non-treated) and 1999 (EGFP-Rab6A, BFA-treated) from a minimum of 41 spheroids per cell line. Data are from two biological replicates. An unpaired Wilcoxon was carried out. *** indicates $p < 0.001$.

EGFP-Rab5A structures, but the number of EGFP-Rab6A structures did increase in a statistically significant manner (S5A Fig), albeit much less than seen following BFA treatment. Measurement of footprint area and texture features revealed a different profile compared to that seen with BFA treatment, suggesting that this approach has the potential to be able to discriminate different phenotypes caused by different drug treatments (S5B Fig).

## Discussion

To date, our understanding of the endomembrane system in cells has almost exclusively come from the use of monolayer models. While these have served us well, it is increasingly being recognised that *in vivo* cells may behave differently, likely as a result of their local environment and the impacts of cell-cell contacts at multiple interfaces. Indeed several studies have shown changes to gene expression in cells when grown in a 3D environment, compared to traditional monolayer cultured cells [4,22–26]. This raises the question of whether studies to date, fully describe how cells are organised, and how they behave. The growth of more complex 3D models of cultured cells in the laboratory environment is not only technically challenging, but poses difficulties from the perspective of high-resolution imaging and image analysis. Nevertheless, given the recent increased interest in 3D models in a number of fields, it seems timely to develop suitable approaches that would enable fundamental questions about intracellular organisation to be addressed [21]. In the work described here, we present a platform that might be used in this way. The first key challenge to overcome was the production of highly

uniform spheroids in a 96-well plate format, compatible with automated imaging methods. Commonly used hydrogel / matrix-based methods produce large numbers of spheroids per well [6,40,41], however, they can be highly variable in size, and are therefore not particularly well suited for screening purposes, or for making comparable measurements between different treatments. Here we demonstrate that micropatterning technology can also support spheroid growth, but bringing high morphological consistency to the spheroid population, thereby facilitating automated imaging, analysis and screening.

To our knowledge, only one previous study has reported growing spheroids using micropatterning technology [30], and in that work, no single cell or subcellular measurements were acquired. Given that a multitude of diseases originate from specific organelles [34–37], we reasoned that developing a 3D cell model that was compatible with automated high-content screening, and that could provide information with subcellular resolution, would be highly valuable. Automated confocal microscopy allowed us to efficiently image several hundred spheroids through their entire volume. We then applied automated image analysis pipelines to the individual cells in the spheroids, allowing us to make various measurements relating to organelle size, morphology and position. While it is important to note that fluorescence is a relatively poor tool for measuring absolute size or distance due to how the captured light disperses from its site of origin, it is still extremely useful for making comparative measurements as required by screening. Our approach revealed that in the HeLa Kyoto cells used in this study, the Golgi apparatus had a median volume of 125 $\mu m^3$ and a median surface area of 500 $\mu m^2$. These values are in line with another study, also utilising fluorescence microscopy, which measured the Golgi volume and surface area in healthy and Alzheimer's disease brain tissue samples. That study reported a Golgi volume of approximately 40 $\mu m^3$ and a surface area of approximately 500 $\mu m^2$ for healthy neocortex cells [42].

For smaller organelles such as endosomes and lysosomes, the limits of resolution in the fluorescence light microscope make it more challenging to obtain accurate measurements of their size and morphology, however, the ability to make comparisons between treatments is still incredibly valuable. For example, a recent study investigating populations of endosomes decorated with various Rab proteins in monolayer-grown non-small cell lung cancer cells measured a more than 50% increase in endosome area following receptor-mediated endocytosis. This increase was driven by Rab4 and Rab5 [43]. Another study developed automated image analysis pipelines using CellProfiler to measure organelle morphology, including endosomes, in monolayer cells. That work measured approximately 35 and 50 early endosomes per cell in HEK293T cells and ECFC cells, respectively [44]. In our work, it was not possible to accurately segment and identify individual endosomes within cells growing as spheroids due to their close physical proximity to each other. However, we were able to estimate their total volume occupied and surface area, parameters which are useful when looking for perturbations in cell organisation. The scenario was similar with respect to lysosomes, which were particularly highly clustered in our cell models. This is certainly one limitation of our overall approach, and other image modalities would be needed to accurately measure number or size of small organelles. Super-resolution imaging methods have been applied in traditional monolayer cells and tumour xenografts to investigate the morphology and size of endosomes in breast cancer cells. Using 3D dSTORM microscopy to obtain volumetric measurements of endosomes, a mean volume of 0.3 $\mu m^3$ was determined [45]. Other studies have used electron or fluorescence microscopy approaches to measure the relative sizes of endosomes [46–49] and lysosomes [50,51]. While super-resolution imaging and electron microscopy are undoubtedly able to provide us with more accurate organelle size and number information, neither can be applied easily to 3D cell models or used in an automated high-throughput screening format. In this regard, our approach deliberately was focused on producing small, uniform spheroids,

fully compatible with automated imaging and analysis. We encountered no difficulties in acquiring high quality image data through depths of up to 150 μm using a 63x/1.15 NA water immersion objective lens, however light scattering and loss of information in spheroids thicker than this would become a limiting factor in terms of being able to accurately quantify subcellular features.

Importantly, we wanted to demonstrate that the 3D cell model platform that we have developed here can be used for quantitative cell-based assays. In monolayer cells, it is common to use transient transfection and expression of GFP-fusions to reveal particular proteins of interest using fluorescence imaging. In spheroids, however, transient transfection is problematic, as invariably only a proportion of the cells in a single assembly will be transfected. To overcome this limitation, we generated cell lines stably expressing certain well-known markers of the endomembrane system, and used these cells to produce spheroids. This allowed us to expand the range of organelles and structures to be studied, and paves the way for their future study using live imaging. Quantitative analysis of the spheroids stably expressing EGFP-Rab5A or EGFP-Rab6A revealed that their overall characteristics were similar to parental cells. We therefore used these spheroids in assays known to perturb endomembrane system function, to test the sensitivity of our quantification methods at the organelle level. BFA is known to inhibit ARF1 GTPase activity by targeting its nucleotide exchange factor GBF1, in turn preventing assembly of the COPI coat on Golgi membranes. This leads to instability of the organelle and the redistribution of lumenal Golgi contents to the ER [52]. This Golgi redistribution assay is commonly used in the field of membrane trafficking to study the Golgi-to-ER retrograde pathway [53–56]. Qualitative changes in the Golgi apparatus were seen in all three cell lines. Using texture analysis tools, we were able to quantify the redistribution of EGFP-Rab6A to a reticular pattern in the cells following BFA treatment, as well as GM130 to distinct punctate structures. These redistributions were consistent with those seen by others in monolayer cells [55–57]. The use of texture analysis tools in 3D cell models, as we demonstrate here, paves the way for conducting large-scale phenotypic screens as used in monolayer cells [58–60]. A second cell-based assay utilising the microtubule depolymerising agent nocodazole, which also results in Golgi fragmentation albeit through a different mechanism, could also be quantified successfully, with the analysis revealing a different trend. This is significant, as it suggests that it will now be possible to use 3D cell models in a high-throughput manner to screen either chemical or RNA interference libraries and detect changes in cell organisation at the subcellular level. One challenge still to overcome is the size and complexity of organelle-level data coming from spheroids grown in a screening-compatible format. Measurements of tens of thousands of individual organelles across large populations of cells means that even very small changes in numerical values can appear statistically significant, however biologically this may not be relevant.

In conclusion, we describe a platform for the investigation of the endomembrane system in cells growing in a 3D environment. We demonstrate that functional / perturbation assays, traditionally used only in monolayer cells, can also be performed in spheroids. We also show that it is possible to obtain high-resolution confocal images of spheroids and carry out volumetric image analysis to make measurements of spheroids, cells and organelles from a single set of image data. We believe that this system has the potential to not only provide a more holistic view of the endomembrane system, but that it could be used in a screening format to help understand the molecular basis of a number of human diseases.

## Supporting information

**S1 Fig. Segmentation of whole spheroids, individual cells and organelles.** Example showing how spheroids were volumetrically analysed using Harmony software. Example segmentations

of A) whole spheroid, B) individual nuclei, C) cytoplasm, D) Golgi (GM130) and E) early endosomes (EEA1). Pseudocolouring is used to highlight individual objects. Scale bar: 50 μm.
(TIF)

**S2 Fig. Quantitative analysis of early endosomes in HeLa Kyoto cells growing as spheroids.** Cells were grown on a CYTOO DC45-P300-FN plate for three days to generate spheroids. The spheroids were fixed and stained with Hoechst 33342 (nuclei), and anti-EEA1 antibodies were used to immunostain the early endosomes. Images were acquired with an Opera Phenix High-Content Screening microscope using a 63x/1.15 NA water immersion objective. A-D) Representative images of the spheroids generated from the various cell lines. Scale bar: 50 μm. E) Mean total endosome volume per cell. F) Mean total endosome surface area per cell. G) Mean number of endosomes per cell. The box plots show per cell measurements indicating the median values and 50 percentiles of the data. Number of cells analysed were 3707 (parental), 2437 (EGFP), 532 (EGFP-Rab5A), and 1304 (EGFP-Rab6A) from a minimum of 12 spheroids per cell line. Data are from three technical replicate wells for each cell line. An unpaired Wilcoxon test was performed. ** indicates $p<0.01$ and *** indicates $p<0.001$.
(TIF)

**S3 Fig. Quantitative analysis of lysosomes in HeLa Kyoto cells growing as spheroids.** Cells were grown on a CYTOO DC45-P300-FN plate for three days to generate spheroids. The spheroids were fixed and stained with Hoechst 33342 (nuclei), and anti-LAMP1 antibodies were used to immunostain the lysosomes. Images were acquired with an Opera Phenix High-Content Screening microscope using a 63x/1.15 NA water immersion objective. A-D) Representative images of the spheroids generated from the various cell lines. Scale bar: 50 μm. E) Mean total lysosome volume per cell. F) Mean total lysosome surface area per cell. G) Mean number of lysosomes per cell. The box plots show per cell measurements indicating the median values and 50 percentiles of the data. Number of cells analysed were 3346 (parental), 2502 (EGFP), 1536 (EGFP-Rab5A), and 1143 (EGFP-Rab6A) from a minimum of 20 spheroids per cell line. Data are from three technical replicate wells for each cell line. An unpaired Wilcoxon test was performed. ns indicates not significant and *** indicates $p<0.001$.
(TIF)

**S4 Fig. Nocodazole-induced Golgi perturbation in spheroids.** After three days of growth in a CYTOO DC45-P300-FN plate, spheroids were treated with 7.5 μM nocodazole for 30 min at 37˚C. Spheroids were fixed and stained with Hoechst 33342 (nuclei), and antibodies were used to immunostain GM130 (Golgi). Images were acquired with an Opera Phenix High-Content Screening microscope with the 63x/1.15 NA water immersion objective. A) Representative images of untreated treated spheroids. Scale bar: 50 μm. B) Representative images of nocodazole treated spheroids. Scale bar: 50 μm. C) Mean Golgi volume for untreated and treated cells grown as spheroids for the various cell lines. D) Number of Golgi fragments for untreated and treated cells grown as spheroids for the various cell lines. The box plots represent per cell measurements. Number of cells analysed were 2923 (parental, non-treated), 2684 (parental, nocodazole-treated), 1994 (EGFP-Rab5A, non-treated), 1951 (EGFP-Rab5A, nocodazole-treated), 2470 (EGFP-Rab6A, non-treated) and 1897 (EGFP-Rab6A, nocodazole-treated) from a minimum 37 spheroids per cell line. Data are from two biological replicates. An unpaired Wilcoxon test was performed. *** indicates $p<0.001$.
(TIF)

**S5 Fig. Quantification of EGFP-Rab5A and EGFP-Rab6A perturbation following nocodazole treatment in spheroids.** Spheroids were grown for three days in a CYTOO DC45-P300-FN plate. Spheroids were treated with 7.5 μM nocodazole for 30 min at 37˚C.

Then spheroids were fixed and stained with Hoechst 33342 (nuclei), and antibodies were used to immunostain GM130 (Golgi). After image acquisition, a volumetric analysis was performed. A) Quantification of the EGFP-Rab5A and EGFP-Rab6A positive structures. B) EGFP-Rab6A footprint area and texture (plane bright and saddle) measurements. The box plots represent per cell measurements. The number of cells analysed were 2016 (EGFP-Rab5A, non-treated), 1957 (EGFP-Rab5A, nocodazole-treated), 2066 (EGFP-Rab6A, non-treated) and 1799 (EGFP-Rab6A, nocodazole-treated) from a minimum of 41 spheroids per cell line. Data are from two biological replicates. An unpaired Wilcoxon test was performed. * indicates $p < 0.05$ and *** indicates $p < 0.001$.
(TIF)

**S1 Table. Harmony analysis pipeline for measuring spheroid, cell and subcellular morphological features.** Analysis pipeline describes the building blocks and the thresholds used to segment spheroids, cells, Golgi, endosomes and lysosomes. The building block in orange 'Select Population' is used only in parental cells to give parental and GFP cells a consistent population name. The building blocks in green were used to select EGFP-expressing cells. The 'Filter Image (2)' and 'Find Image Region (3)' building blocks provide endosome (light blue) and lysosome (purple) specific thresholds.
(PDF)

**S2 Table. Harmony analysis pipeline for measuring morphological features of the Golgi in non-treated control cells in the BFA and nocodazole assays.** Analysis pipeline describes the building blocks and the thresholds used to segment spheroids, cells and Golgi. The building block in orange 'Select Population' is used only in parental cells to give parental and GFP cells a consistent population name. The building blocks in green were used to select EGFP-expressing cells.
(PDF)

**S3 Table. Harmony analysis pipeline for measuring morphological features of the Golgi in treated cells in the BFA and nocodazole assays.** Analysis pipeline describes the building blocks and the thresholds used to segment spheroids, cells and Golgi. The building block in orange 'Select Population' is used only in parental cells to give parental and GFP cells a consistent population name. The building blocks in green were used to select EGFP-expressing cells.
(PDF)

**S4 Table. Harmony analysis pipeline for measuring number of EGFP-Rab5A positive structures in the BFA and nocodazole assays.** Analysis pipeline describes the building blocks and the thresholds used to segment spheroids, cells and EGFP-Rab5A-positive structures.
(PDF)

**S5 Table. Harmony analysis pipeline for measuring EGFP-Rab6A features in the BFA and nocodazole assay.** Analysis pipeline describes the building blocks and the thresholds used to segment spheroids, cells and EGFP-Rab6A-positive structures. Building blocks 'Calculate Image' and 'Find Image region (2)' contain specific values for non-treated, BFA (purple)- and nocodazole-treated (light blue) EGFP-Rab6A cells.
(PDF)

**S6 Table. Harmony analysis pipeline for identifying spheroids for the PreciScan function of the Opera Phenix microscope.** Analysis pipeline describes the building blocks and thresholds used to segment spheroids as well as the building block 'Determine Well Layout' for the rescan with the 63x water immersion objective.
(PDF)

**S7 Table. Harmony Analysis Pipeline for measuring morphological features of spheroids from a single plane.** Analysis pipeline describes the building blocks and thresholds used to segment spheroids as well as what morphological features were measured.
(PDF)

## Acknowledgments

The authors thank all members of the lab for their input and helpful discussions on image analysis. We also thank Claire Gormley for advice on statistical analysis.

## Author Contributions

**Conceptualization:** Margaritha M. Mysior, Jeremy C. Simpson.

**Formal analysis:** Margaritha M. Mysior.

**Funding acquisition:** Margaritha M. Mysior, Jeremy C. Simpson.

**Investigation:** Margaritha M. Mysior.

**Methodology:** Margaritha M. Mysior, Jeremy C. Simpson.

**Project administration:** Jeremy C. Simpson.

**Supervision:** Jeremy C. Simpson.

**Visualization:** Margaritha M. Mysior.

**Writing – original draft:** Margaritha M. Mysior, Jeremy C. Simpson.

**Writing – review & editing:** Margaritha M. Mysior, Jeremy C. Simpson.

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
