## [Decision Letter · Decision Letter 0]

31 May 2024

PONE-D-24-15969An automated high-content screening and assay platform for the analysis of spheroids at subcellular resolutionPLOS ONE

Dear Dr. Simpson,

Thank you for submitting your manuscript to PLOS ONE. After careful consideration, we feel that it has merit but does not fully meet PLOS ONE’s publication criteria as it currently stands. Therefore, we invite you to submit a revised version of the manuscript that addresses the points raised during the review process.

The reviewers have provided a careful and thorough evaluation of your manuscript, and report a considerable list of 'minor' issues that would need to be well-addressed before the manuscript would be acceptable for publication. The reviewers seek to have a rewrite that would better define whether the manuscript is a technical report or a research report; currently it is a mix. Reviewer 1 seeks answers to multiple technical details (1-6) for the plate processing, optical imaging issues, segmentation of organelles, replicates. Reviewer 2 (1, 2a/b) seeks information on the nature of the organoids (a natural structure, or an unusual association of packed cells; explanation of the diffrences in 2D vs 3D responses?), and also if the manuscript is intended to be a technical report, further detail is requested and included in supplemental for the interested readership, followed by a list of minor issues (1-11) that should be addressible. 

We look forward to receiving your revised manuscript.

Kind regards,

Michael A. Mancini

Academic Editor

PLOS ONE

Journal Requirements:

"MMM is supported by the Government of Ireland Postdoctoral Fellowship from the Irish Research Council (GOIPD/2023/1135)"

"The authors thank all members of the lab for their input and helpful discussions. MMM is supported by the Government of Ireland Postdoctoral Fellowship from the Irish Research Council (GOIPD/2023/1135). The JCS Laboratory is supported by the UCD College of Science. "

"MMM is supported by the Government of Ireland Postdoctoral Fellowship from the Irish Research Council (GOIPD/2023/1135)"

5. In this instance it seems there may be acceptable restrictions in place that prevent the public sharing of your minimal data. However, in line with our goal of ensuring long-term data availability to all interested researchers, PLOS’ Data Policy states that authors cannot be the sole named individuals responsible for ensuring data access (http://journals.plos.org/plosone/s/data-availability#loc-acceptable-data-sharing-methods).

Reviewers' comments:

Reviewer's Responses to Questions

**Comments to the Author**

1. Is the manuscript technically sound, and do the data support the conclusions?

Reviewer #1: Partly

Reviewer #2: Yes

2. Has the statistical analysis been performed appropriately and rigorously? 

Reviewer #1: No

Reviewer #2: Yes

3. Have the authors made all data underlying the findings in their manuscript fully available?

Reviewer #1: No

Reviewer #2: Yes

4. Is the manuscript presented in an intelligible fashion and written in standard English?

Reviewer #1: Yes

Reviewer #2: Yes

5. Review Comments to the Author

Reviewer #1: The study by Mysior and Simpson is valuable for the 3D HTM/HCA community but lacks many details that would be required to reproduce and/or scale up the approach, including the primary data. specifically, the following questions should be addressed. Plus a general answer: has the process been tested on different cell models or only in the HeLa subclone described?

1. it would be useful to have more details about the plate processing - i.e., volumes, how are the plate handled (manual, robotic)

2. were different antibody incubation times and/or triton incubation times tested?

3. it is unclear if segmentation was performed on each 2D plane and then somewhat concatenated or as true 3D segmentation. why was the cytoplasm segmented off a "non specific" hoechst signal and not GFP?

4. more details are needed to understand why and how filters were sued to segment organelles and how thresholds were chosen

5. what is a little problematic is that somehow the authors calculate (estimate?) volumes but it is not explained how and the imaging conditions described clearly show that optical section is too big to be at Nyquist, hence the volume will not be precise. also, there is no mention of the obvious z distortion and if/how it was compensated in the calculations

6. indication of replicates (technical and biological) should be clearly spelled out in the methods section

7. the statement "Single cell data could not be statistically analysed with standard statistical tests due to the high number of data values" is unacceptable as the number of single cell data values is not very high and single cell analysis is essential if the effort to segment single cells is implemented

8. what is the fraction of successful spheroid formed over total seeded?

9. in Figure 2 the scale bar seems wrong, especially when compared to the other figures

Reviewer #2: In the manuscript titled “An automated high-content screening and assay platform for the analysis of spheroids at subcellular resolution” the authors present the development of a technique to generate a large number of arrayed relatively uniform spheroids using micropatterned multi-well plates. The authors describe how these samples can be imaged using an automated high-resolution image cytometer and analyzed at the spheroid-, cell-, and subcellular-level using an algorithm developed within the Harmony software platform. Finally, they demonstrate how the assay can detect changes induced by inhibitors that affect the Golgi apparatus or microtubules. The strength of the manuscript is the presented techniques address an ongoing need in the research community to efficiently generate and analyze more representative 3D samples. The key weakness is the limited characterization of the quality of generated spheroid. Although the manuscript would be significantly strengthened if this weakness was addressed, the work presented in the manuscript is well written and scientifically sound and merits publication in PLOS One.

Strengths

1) The authors introduce a method to generate large numbers of relatively uniform spheroids. If proven broadly applicable, this would advance the ability to perform HTS using spheroid samples.

2) The authors demonstrate how tools within the Harmony software platform can be used to achieve sub-cellular analysis resolution.

Weakness

1) It is unclear if this technique is producing quality spheroids with cells tightly interacting with each other or simply cell aggregates centered on the micropatterned ‘well’. The manuscript would be strengthened significantly if efforts were made to assess the quality of the spheroids produced by the micropattern technique. Alternatively, authors could demonstrate how responses differed in the 3D spheroid vs the 2D culture of the cell lines used in the study, which would indicate biological differences associated with the 3D samples.

2) The focus of the manuscript is somewhat unclear.

2a) It appears written primarily as a technique/assay development paper. As such, further detail on the methods/analysis algorithm used could be provided. The optimization work would be important to share in supplemental material. Importantly, it should be made clear if the Harmony algorithm will be made/are available to the research community. In addition, metrics describing assay quality (for example, Z’-prime values) should be calculated and provided.

2b) The authors devote significant space describing the importance of using 3D cultures/spheroids to generate more representative samples to study cell physiology compared to 2D cultures, however, no data was presented to suggest the current assay captures that difference. The advantage of 3D spheroids are intended to be the focus of the manuscript, comparative data with 2D cultures should be included.

Other Comments

1. Line 73 – Missing punctuation/incomplete sentence

2. Line 165 – Methods list several objectives used, but it is unclear in the manuscript where each (if all) were used.

3. Line 166 – Typo “63x/1.1.15NA”.

4. Line 189 – More accurate to state that statistical analysis of single cell data is not presented due to the large number of samples generating significant differences with limited/unclear biological meaning.

5. Line 206-207 – As an assay development paper, it might be useful to describe the optimization process in greater detail for others attempting similar assays using alternative cell types.

6. Figure 1 Legend – Would be good to indicate if displayed image represents a single field or a stitched collection of fields.

7. Line 344-345 – For readers not familiar with the Harmony software, it would be good to provide the type of texture being measured by the ‘plane bright’ and ‘saddle’ texture components.

8. The authors should comment about any issues with photobleaching when collecting a large z-stack at 100% laser power and image exposure times up to 500 msec.

9. It would be informative if authors provided an average scanning time per plate/per well to collect the low and high resolution images.

10. A key limitation of high NA objectives is a limited working depth. While this did not impact the current study due to relatively small spheroids generated using the microprinting technique, it would be good if the authors commented on how this might impact broader application of the methods to other types of

11. Scale Bar length vs spheroid shown in figures. Figure 1D indicates an average spheroid width ranging between ~80-90 um. However, at a scale bar length of 50 um, it appears many of the spheroids shown at higher resolution in figures are approaching ~150 um width, falling within in the far extreme range observed at 20X. Confirm that scale bar length is correct, or it would be good to state in the text/figure legend that especially large spheroids were chosen to allow easier visualization of the phenotype/masking.

6. PLOS authors have the option to publish the peer review history of their article (what does this mean?). If published, this will include your full peer review and any attached files.

Reviewer #1: No

Reviewer #2: **Yes: **Adam T. Szafran

---

## [Author Response · Author response to Decision Letter 0]

12 Aug 2024

Academic Editor Comments:

This has been checked and our manuscript conforms to the correct style.

We are not sure which elements ‘do not match’, however we believe that we have added the correct information to the relevant boxes on resubmission.

"MMM is supported by the Government of Ireland Postdoctoral Fellowship from the Irish Research Council (GOIPD/2023/1135)”. Please state what role the funders took in the study. If the funders had no role, please state: "The funders had no role in study design, data collection and analysis, decision to publish, or preparation of the manuscript."

We have added this statement, as indeed the funders played no role in the study itself.

"The authors thank all members of the lab for their input and helpful discussions. MMM is supported by the Government of Ireland Postdoctoral Fellowship from the Irish Research Council (GOIPD/2023/1135). The JCS Laboratory is supported by the UCD College of Science. " We note that you have provided funding information that is not currently declared in your Funding Statement. However, funding information should not appear in the Acknowledgments section or other areas of your manuscript. 

We have removed the funding information from the Acknowledgements as requested.

5. In this instance it seems there may be acceptable restrictions in place that prevent the public sharing of your minimal data. However, in line with our goal of ensuring long-term data availability to all interested researchers, PLOS’ Data Policy states that authors cannot be the sole named individuals responsible for ensuring data access.

We can provide contact information for UCD Research, the unit that oversees all research activities in the university, however in fact the PIs are a more stable form of contact than the professional staff who work in UCD Research. Also to note, UCD Research takes no responsibility for holding the research data of individual PIs. In UCD PIs are obliged to hold their own data in a responsible manner, and all PIs must complete a research integrity course that addresses this key point, and then comply with it.

6. Please review your reference list to ensure that it is complete and correct.

We have reviewed our reference list and believe it to be accurate. None of the cited works have been retracted.

Reviewer #1: 

We thank the reviewer for their time reviewing our manuscript and we appreciate the constructive feedback. We have read the feedback carefully and have implemented their suggestions, which we believe address all the concerns raised. On the headline comment that our original manuscript lacks details that would allow others to reproduce the approach, in the revised manuscript we now provide significantly more methodological detail, and importantly we include the full image analysis pipelines used in all experiments.

The study by Mysior and Simpson is valuable for the 3D HTM/HCA community but lacks many details that would be required to reproduce and/or scale up the approach, including the primary data. specifically, the following questions should be addressed. Plus a general answer: has the process been tested on different cell models or only in the HeLa subclone described?

In the original optimisation experiments carried out, we worked with both HeLa Kyoto and HEK293 cells. The HeLa Kyoto cells formed spheroids in a more consistent manner than the HEK293 cells, and so these were taken forward in the study. Similar micropatterned plates have been shown (in a publication from researchers at CYTOO) to support spheroid growth based on HCT-116, T-47D, MCF-7, MDA-MB-231, A549, HT-29 and HeLa cells, although in that paper no single cell or subcellular information was acquired. These details have been added to the revised manuscript (lines 234-238). 

1. it would be useful to have more details about the plate processing - i.e., volumes, how are the plate handled (manual, robotic)

We have now provided an expanded Materials and Methods section giving these additional details (lines 143-144, 146-147, 149-150, 153-157).

2. were different antibody incubation times and/or triton incubation times tested?

The antibodies used in this particular study are well known to us and required little optimisation. However, we have added a note in the revised manuscript suggesting that optimisation of other antibodies is required (lines 153-154). 

3. it is unclear if segmentation was performed on each 2D plane and then somewhat concatenated or as true 3D segmentation. why was the cytoplasm segmented off a "non specific" hoechst signal and not GFP?

The 3D segmentation algorithms with Harmony calculate a 3D image based on the selected stack planes. The entire 3D image (image cube) is analysed instead of using single planes (or maximum projection). This information has now been added to the image analysis section of the Materials and Methods (lines 189-191). The use of residual Hoechst in the cytoplasm is a commonly used approach in image analysis to identify entire cells. It provides two key advantages in studies such as ours. Firstly, it means that only one colour channel is needed to segment individual cells and nuclei, leaving three additional channels available for more critical information (one GFP-channel and two antibody stains). Secondly, it also means that cell lines not expressing GFP-tagged proteins (for example the parental cells used in this study) can still be identified and segmented in exactly the same way as the cell lines stably expressing GFP-tagged proteins. A sentence has been added to the image analysis section acknowledging that a plasma membrane stain can also be used to segment cells and the cytoplasm (lines 189-191). 

4. more details are needed to understand why and how filters were used to segment organelles and how thresholds were chosen

Given that the focus of this paper is more methodological, and that we intended to provide sufficient details for others to be able to replicate this approach, in the revised manuscript, we now provide the complete image analysis pipelines as tables (Tables S1 to S7) in the Appendix detailing which building blocks, filters and threshold values were used to segment spheroids, cells and organelles. Information about the analysis pipelines has been added to the Material and Methods section (lines 205-206).

5. what is a little problematic is that somehow the authors calculate (estimate?) volumes but it is not explained how and the imaging conditions described clearly show that optical section is too big to be at Nyquist, hence the volume will not be precise. also, there is no mention of the obvious z distortion and if/how it was compensated in the calculations

We chose this optical sectioning distance as a compromise to reduce imaging times and data volumes, given that our methodological approach is designed to be used in a screening format with potentially large numbers of plates. For the high-magnification images of spheroids a slice interval of 1 um was selected, which is indeed double the Nyquist distance (0.5 um) for this particular objective. We added a sentence in the Methods section clarifying this point (lines 177-180). However, throughout the manuscript we repeatedly acknowledge that volumetric measurements made using any form of fluorescence should be treated with caution (for example lines 463-466, 473-476, 493-495) and that rather that these data are better used in a comparative way between samples. 

6. indication of replicates (technical and biological) should be clearly spelled out in the methods section

The specific number of replicates varies slightly between each experiment. In the original manuscript, this information was already provided in each figure legend. However, we acknowledge that this information was missing from Figure 1, and this has now been added. 

7. the statement "Single cell data could not be statistically analysed with standard statistical tests due to the high number of data values" is unacceptable as the number of single cell data values is not very high and single cell analysis is essential if the effort to segment single cells is implemented

Statistical analysis has been performed for the data in Figures 3-5 and S2-S5. The figures have been modified to reflect this and information on how the statistical analysis was performed has been added (lines 211-214) and is in the relevant figure legends. It is important to note, however, that when comparing features for thousands of individual organelles from hundreds of cells, even small differences between values result in extremely small p-values that may be misleading. We have discussed this with an expert statistician and taken advice on how to analyse very large data sets such as this. This point is also now included in the Discussion section (lines 526-530).

8. what is the fraction of successful spheroid formed over total seeded?

In a typical experiment, approximately 90 % of the micropatterns contained cells 1h to 2h after cell seeding. From this, 60-80 % of the micropatterns eventually support spheroid formation and development after 3 days of growth. We added this information into the revised manuscript (lines 241-243).

9. in Figure 2 the scale bar seems wrong, especially when compared to the other figures

We thank the reviewer for spotting this. Indeed the scale bar was incorrectly annotated on the figure. The scale bar has been corrected, and a new Figure 2 uploaded.

Reviewer #2: 

We thank the reviewer for their time reviewing our manuscript and we appreciate that this reviewer notes that the work merits publication in PLOS One. We have read the feedback carefully and have implemented their suggestions, which we believe address all the concerns raised. 

In the manuscript titled “An automated high-content screening and assay platform for the analysis of spheroids at subcellular resolution” the authors present the development of a technique to generate a large number of arrayed relatively uniform spheroids using micropatterned multi-well plates. The authors describe how these samples can be imaged using an automated high-resolution image cytometer and analyzed at the spheroid-, cell-, and subcellular-level using an algorithm developed within the Harmony software platform. Finally, they demonstrate how the assay can detect changes induced by inhibitors that affect the Golgi apparatus or microtubules. The strength of the manuscript is the presented techniques address an ongoing need in the research community to efficiently generate and analyze more representative 3D samples. The key weakness is the limited characterization of the quality of generated spheroid. Although the manuscript would be significantly strengthened if this weakness was addressed, the work presented in the manuscript is well written and scientifically sound and merits publication in PLOS One.

Strengths

1) The authors introduce a method to generate large numbers of relatively uniform spheroids. If proven broadly applicable, this would advance the ability to perform HTS using spheroid samples.

2) The authors demonstrate how tools within the Harmony software platform can be used to achieve sub-cellular analysis resolution.

Weakness

1) It is unclear if this technique is producing quality spheroids with cells tightly interacting with each other or simply cell aggregates centered on the micropatterned ‘well’. The manuscript would be strengthened significantly if efforts were made to assess the quality of the spheroids produced by the micropattern technique. Alternatively, authors could demonstrate how responses differed in the 3D spheroid vs the 2D culture of the cell lines used in the study, which would indicate biological differences associated with the 3D samples.

The focus of this manuscript is technical in nature (rather than presenting biological results from a screen, or making a direct comparison between 2D- and 3D-grown cells), detailing an approach that we believe has the potential to facilitate high-resolution imaging from a small spheroid model in an automated screening format. Our initial optimisation experiments and associated calculations suggest that the spheroids all form from the initial cells that are deposited on the micropattern directly following cell seeding. Unfortunately, we are a little unsure what ‘quality’ measure this reviewer is requesting. We believe that this is the most detailed quantitative characterisation of spheroids presented to date, providing information on sphericity, volume, surface area, and number of nuclei. Interestingly, these values are comparable to those measured in small spheroids generated using more ‘conventional’ production methods, for example growing them in extracellular matrix (ECM) basement membrane material (Ref. no. 41). This suggests that their characteristics (quality) are similar to spheroids formed using established methods. 

2) The focus of the manuscript is somewhat unclear.

We have made this manuscript more technical / methodological in focus, by providing more detail in the Materials and Methods section and Appendices (lines 140-157, 175-185, 195-206, 233-238). Importantly, we now provide the detailed image analysis pipelines used to generate the various quantitative results, enabling others to replicate our experiments.

2a) It appears written primarily as a technique/assay development paper. As such, further detail on the methods/analysis algorithm used could be provided. The optimization work would be important to share in supplemental material. Importantly, it should be made clear if the Harmony algorithm will be made/are available to the research community. In addition, metrics describing assay quality (for example, Z’-prime values) should be calculated and provided.

Please refer to the response above. We have made all the analysis pipelines available as tables in the Appendix. The Harmony software is proprietary software from Revvity and we are unable to distribute it. However, in other work we have previously shown how image analysis pipelines generated in Harmony can be replicated in the open-source software Cell Profiler (Garcia-Pardo et al., 2021; PMID:34496765). It is not possible to calculate Z’ values for the data shown in this work. Z’ values are used in screens to compare experimental values to positive and negative controls, and thereby determine the screen quality. However, in the work that we present, there are no positive controls as such; rather the approach details how strengths of phenotypes can be compared. It should also be noted that Z’ values can only be applied to data that show a normal distribution; this also does not apply here. 

2b) The authors devote significant space describing the importance of using 3D cultures/spheroids to generate more representative samples to study cell physiology compared to 2D cultures, however, no data was presented to suggest the current assay captures that difference. The advantage of 3D spheroids are intended to be the focus of the manuscript, comparative data with 2D cultures should be included.

In the revised manuscript, we have added significantly more methodological information to make this a technical paper (lines 140-157, 175-185, 195-206, 233-238). Carrying out a comparative screen with monolayer cells would be a manuscript in its own right and is far beyond the scope of this work. 

Other Comments

1. Line 73 – Missing punctuation/incomplete sentence

We apologise for this error, caused by a change in formatting just prior to submission. The missing punctuation has been added and the incomplete sentence finished (see line 77).

2. Line 165 – Methods list several objectives used, but it is unclear in the manuscript where each (if all) were used.

Each figure legend details the specific objective used to generate the corresponding image data. This information c

---

## [Decision Letter · Decision Letter 1]

9 Sep 2024

PONE-D-24-15969R1An automated high-content screening and assay platform for the analysis of spheroids at subcellular resolutionPLOS ONE

Dear Dr. Simpson,

Thank you for submitting your manuscript to PLOS ONE. After careful consideration, we feel that it has merit but does not fully meet PLOS ONE’s publication criteria as it currently stands. Therefore, we invite you to submit a revised version of the manuscript that addresses the points raised during the review process.

We look forward to receiving your revised manuscript.

Kind regards,

Michael A. Mancini

Academic Editor

PLOS ONE

Journal Requirements:

Additional Editor Comments:

Dear authors:

The manuscript has been markedly improved. I expect that the minor, but the very useful concerns of Reviewer 2 would be straightforward to respond to, which would result in the manuscript becoming acceptable.

Reviewers' comments:

Reviewer's Responses to Questions

**Comments to the Author**

1. If the authors have adequately addressed your comments raised in a previous round of review and you feel that this manuscript is now acceptable for publication, you may indicate that here to bypass the “Comments to the Author” section, enter your conflict of interest statement in the “Confidential to Editor” section, and submit your "Accept" recommendation.

Reviewer #1: All comments have been addressed

Reviewer #2: All comments have been addressed

2. Is the manuscript technically sound, and do the data support the conclusions?

Reviewer #1: Yes

Reviewer #2: Yes

3. Has the statistical analysis been performed appropriately and rigorously? 

Reviewer #1: Yes

Reviewer #2: Yes

4. Have the authors made all data underlying the findings in their manuscript fully available?

Reviewer #1: Yes

Reviewer #2: Yes

5. Is the manuscript presented in an intelligible fashion and written in standard English?

Reviewer #1: Yes

Reviewer #2: Yes

6. Review Comments to the Author

Reviewer #1: we thank the authors for addressing all the points that were discussed. the manuscript now contains all the information needed for publication

Reviewer #2: With the revisions and added technical details, the authors have address the primary weakness and clearly made the paper focused on the technical protocol for analyzing the 3D spheroids. The previous strengths remain. The manuscript is suitable for publication in PLOS One with some minor revisions.

Minor Comments:

[Line 189] The authors state spheroids were segmented using the A647 secondary antibody signal. For any significant screening project, the use of antibodies would significantly increase the cost per well. Therefore, it would be beneficial if the authors could comment if similar spheroid region segmentation is possible using either the DNA dye or GFP signal (when present).

[General comment] The authors describe the developed tools as useful for screening, however, more effort could be made to describe how much variation observed between replicate samples. For example, well-to-well variation in the average number of detected objects per cell or per spheroid.

[Line 282] The use of the phrase "a high degree of accuracy" suggests the use of a metric that quantifies segmentation quality compared to an annotated ground truth set of images (i.e. IoU, F1, etc analysis). If no such comparison was performed. it may be better to make a statement stating that the assessment of accuracy was qualitative and not quantitative.

[General Comment] Charts appear low resolution in the PDF. This is not an issue with the supplemental figures.

[General Comment] Instances where the authors' description of the data does not match what the figure shows. For example, authors describe figure 5A data (line 367) as 'no detectable difference in the number of structures', yet, the figure indicates a significant difference. This is a carry-over of the revision now containing the statistical analysis of single-cell data, but the language used need to be corrected.

[Line 732] The figure legend does not indicate the meaning of the colors used. Are the individual objects pseudo-colored?

7. PLOS authors have the option to publish the peer review history of their article (what does this mean?). If published, this will include your full peer review and any attached files.

Reviewer #1: No

Reviewer #2: **Yes: **Adam T. Szafran, MD/PhD

---

## [Author Response · Author response to Decision Letter 1]

11 Sep 2024

Reviewer #2: 

We now address the additional queries mentioned by this reviewer, which we note several of which were not raised in their original assessment of the manuscript.

[Line 189] The authors state spheroids were segmented using the A647 secondary antibody signal. For any significant screening project, the use of antibodies would significantly increase the cost per well. Therefore, it would be beneficial if the authors could comment if similar spheroid region segmentation is possible using either the DNA dye or GFP signal (when present).

In lines 189-191, we already stated in the previous version of the manuscript that spheroid-level segmentation based on either a plasma membrane stain or the Hoechst 33342 staining is possible. We refer the reviewer to lines 189-191 both in this version and the previous manuscript version.

[General comment] The authors describe the developed tools as useful for screening, however, more effort could be made to describe how much variation observed between replicate samples. For example, well-to-well variation in the average number of detected objects per cell or per spheroid.

This is an additional query not raised in the original review of the manuscript. We have analysed the numbers of spheroids detected between replicate wells and determined the coefficient of variance to be in the order of 5% between replicate wells. A comment to this effect has now been added (see lines 258-260).

[Line 282] The use of the phrase "a high degree of accuracy" suggests the use of a metric that quantifies segmentation quality compared to an annotated ground truth set of images (i.e. IoU, F1, etc analysis). If no such comparison was performed. it may be better to make a statement stating that the assessment of accuracy was qualitative and not quantitative.

This is an additional query not raised in the original review of the manuscript. The phrase “a high degree of accuracy” has been removed (see lines 284-285).

[General Comment] Charts appear low resolution in the PDF. This is not an issue with the supplemental figures.

All of the graphical data, throughout the manuscript, have been prepared in an identical manner and following the journal guidelines (300 dpi). We suggest to the reviewer that the low resolution of the graphs in the pdf is a consequence of how the pdf builder on the journal’s website assembles the document in a compressed format for peer review, and so this would not be expected to be an issue in the final published version.

[General Comment] Instances where the authors' description of the data does not match what the figure shows. For example, authors describe figure 5A data (line 367) as 'no detectable difference in the number of structures', yet, the figure indicates a significant difference. This is a carry-over of the revision now containing the statistical analysis of single-cell data, but the language used need to be corrected.

The choice of language to describe these changes has now been modified (see lines 311, 339-340, 363, 369-370, 413, 418). We also again refer the reviewer to lines 529-533 of the manuscript where we discuss the use and limitations of statistical tools when analysing large numbers of individual object data in high-content screens. 

[Line 732] The figure legend does not indicate the meaning of the colors used. Are the individual objects pseudo-colored?

This is an additional query not raised in the original review of the manuscript. A sentence clarifying this has been added to the figure legend S1 (see line 738).

---

## [Editor Report · Decision Letter 2]

29 Sep 2024

An automated high-content screening and assay platform for the analysis of spheroids at subcellular resolution

PONE-D-24-15969R2

Dear Dr. Simpson,

We’re pleased to inform you that your manuscript has been judged scientifically suitable for publication and will be formally accepted for publication once it meets all outstanding technical requirements.

Kind regards,

Michael A. Mancini

Academic Editor

PLOS ONE
---

## [Editor Report · Acceptance letter]

7 Oct 2024

PONE-D-24-15969R2 

PLOS ONE

Dear Dr. Simpson, 

I'm pleased to inform you that your manuscript has been deemed suitable for publication in PLOS ONE. Congratulations! Your manuscript is now being handed over to our production team.

Kind regards, 

on behalf of

Dr. Michael A. Mancini 

Academic Editor

PLOS ONE